# “*We’re Home Now*”: How a Rehousing Intervention Shapes the Mental Well-Being of Inuit Adults in Nunavut, Canada

**DOI:** 10.3390/ijerph19116432

**Published:** 2022-05-25

**Authors:** Karine Perreault, Josée Lapalme, Louise Potvin, Mylène Riva

**Affiliations:** 1École de Santé Publique, Université de Montréal, Montréal, QC H3N 1X9, Canada; j.lapalme@umontreal.ca (J.L.); louise.potvin@umontreal.ca (L.P.); 2Centre de Recherche en Santé Publique, CIUSSS du Centre-Sud-de-l’Île-de-Montréal, Université de Montréal, Montréal, QC H3L 1M3, Canada; 3École de Psychoéducation, Université de Montréal, Montréal, QC H2V 2S9, Canada; 4Institute for Health and Social Policy, Department of Geography, McGill University, Montreal, QC H3A 0B9, Canada

**Keywords:** housing construction, social housing, rehousing intervention, Indigenous, Inuit, mental health, health promotion, social determinant of health, Nunavut

## Abstract

This study explores the ways in which a rehousing intervention shapes the mental well-being of Inuit adults living in Nunavut, Canada, where the prevalence of core housing need is four times the national average. More specifically, it compares the housing experiences of participants who were rehoused in a newly built public housing unit, to the experiences of participants on the public housing waitlist. The study was developed in collaboration with organizations based in Nunavut and Nunavik. Semi-structured interviews were transcribed, and a deductive-inductive thematic analysis was performed based on Gidden’s concept of ontological security, and Inuit-specific mental health conceptualization. Twenty-five Inuit adults participated (11 rehoused, 14 waitlist). Three themes were identified to describe how the subjective housing experiences of participants improved their mental well-being after rehousing: (1) refuge creation; (2) self-determination and increased control; (3) improved family dynamics and identity repair. Implicit to these themes are the contrasting housing experiences of participants on the waitlist. Construction initiatives that increase public housing stock and address gaps in the housing continuum across Inuit regions could promote well-being at a population level. However, larger socio-economic problems facing Inuit may hamper beneficial processes stemming from such interventions.

## 1. Introduction

Inuit regions in the Canadian Arctic, collectively known as Inuit Nunangat (Figure 1, https://www.itk.ca/inuit-nunangat-map/ (accessed on 20 May 2022)), are experiencing a persistent and growing housing crisis which has been characterized as “*one of the most significant public health emergencies in Canada*” [1]. Indeed, important housing gaps between Inuit and non-Indigenous groups contribute to the social inequalities in health observed between these two groups in Canada [2]. In 2016, 40% of Inuit were considered in core housing need, based on housing adequacy, suitability (crowding), affordability, and availability standards (Appendix A), compared to 10% of non-Indigenous Canadians [3]. National and regional health surveys have reported significant associations between poor housing quality, household overcrowding and mental distress among Inuit [4,5,6]. However, the lived experiences of such housing conditions and their implications for mental well-being have rarely been explored from an Inuit perspective, with few exceptions [7,8,9] and never in the context of a housing intervention. Between 2015 and 2018, over 300 public housing units were constructed in Nunavut [10,11,12], one of the four Inuit regions in Canada. To put this figure into perspective, in 2009–2010, it was estimated that 3580 new dwellings were required to house people living in substandard housing (in need of major repairs and/or overcrowded) in Nunavut and who would move out if more housing were available in their community. Far from reaching the population’s housing needs ([13], p. 26), these newly built units nevertheless provided the opportunity for several Inuit families to be rehoused and improved their housing circumstances [14].

This article aims to explore the ways in which Inuit adults living in Nunavut who had been rehoused, and Inuit who applied to the waitlist for public housing and were still living in substandard conditions, perceived the rehousing intervention to affect, or to potentially affect, their mental well-being. The study was conducted in a remote and isolated community of Nunavut, the northernmost Canadian territory located above the 60th parallel, where none of the 25 communities have road access. Findings underscore the important benefits to mental well-being for participants who have been rehoused, especially in contrast to those on a waitlist. They speak to the transformative and health-promoting potential of housing interventions devoted to addressing the housing crisis in Nunavut.

## 2. Literature Review

### 2.1. Housing, Mental Well-Being, and Ontological Security

Although the links between poor housing quality, overcrowding, and poor physical health outcomes have been demonstrated [15], there is a knowledge gap when it comes to understanding the impacts of housing on mental health [16], and even more so regarding the health impacts of subjective housing experiences [17]. To respond to this gap, we turned to the ontological security theory [18] as it provides theoretical and empirical starting points to reflect on how housing shapes health, particularly mental health.

Stated simply, ontological security is a sense of confidence and trust in the world as it appears to be [19]. It is a security of being, derived from the confidence that most human beings have in the continuity of their self-identity and in the constancy of their social and material environments [19]. In Gidden’s view, the capacity to trust is necessary for emotional and psychological well-being for all individuals in all societies [18,20]. The realities from which ontological security is acquired and maintained are certainly culturally specific, but Giddens asserts that no human can function normally without ontological security. We return later to the relevance of using this theory in relation to the Inuit conception of mental well-being, and in the context of a rehousing intervention in the Inuit homeland.

The notion of ontological security has been developed specifically in relation to the home environment, in that “home is where people feel in control of their environment, free from surveillance, free to be themselves and at ease, in the deepest psychological sense, in a world that might at times be experienced as threatening and uncontrollable” [21]. More specifically, Dupuis and Thorns (1998) showed that the home can be a source of ontological security when the following four conditions are met: (i) home is the site of consistency in the social and material environment; (ii) home is a spatial context in which the day-to-day routines of human existence are performed; (iii) home is a site where people feel most in control of their lives because they feel free from the surveillance that is part of the contemporary world; (iv) home is a secure base around which identities are constructed.

Giddens’ original meaning of ontological security can be understood as a “normal and latent state” (as opposed to the “abnormal” state of mental illness), remaining in the subconscious unless threatened [18]. Although its subconscious nature has been challenged, if we accept the idea of its latent nature, making the demonstration of its existence through empirical evidence remains a difficult task. Kearns et al. (2000), drawing on the work of Dupuis and Thorns (1998), developed the concept of ‘ontological security from the home’ in a sense slightly departing from Gidden’s views, representing the immediate, conscious, yet subjective housing experiences conducive to well-being. The work of Kearns et al. (2000) was also instrumental in highlighting the social significance of homes and the importance of positive family relationships for feelings of trust and well-being to emerge from one’s housing experiences [22].

In the last two decades, the notion of ontological security has been used to explain important functions of the home and their impacts on individuals’ health and well-being. The argument put forward, in essence, is that the home can serve as a medium for mental health promotion, when it provides a secure and stable tenancy (constancy), along with living conditions that promote individuals’ agency (autonomy and control) [17,23,24,25,26,27,28]. Home also relates to ontological security and well-being when experienced as a place of social connections and affection, which in turn support positive identity construction [23,24,25,27].

It must be acknowledged that housing scholars have been criticized for largely ignoring the negative impacts of housing on ontological (in)security [29]. One way to overcome these limitations is to carefully consider the wider context in which home has developed its significance as a source of ontological security, and to explore how the interactions with the context can potentially hamper the anticipated benefits or trigger unexpected effects. The geographic, economic, and historical contexts in which people evolve are understood socially in the creation of a place and form the foundation of local culture [30]. Kearns et al. (2000) argue that lived experiences of the home are heavily influenced by what has been termed “local cultures of home living”—the way in which housing is locally consumed and inhabited in its residential context.

In this article, we use the concept of “ontological security from the home” [19,22] to better understand the immediate and conscious mechanisms by which lived experiences of housing influence perceived well-being, in the context of the housing crisis in Nunavut.

### 2.2. The Housing Context in Inuit Nunangat and Its Impact on Mental Well-Being

Inuit have inhabited the Arctic regions of Canada and circumpolar countries for at least 5000 years [31]. They used to be nomadic, living in dwellings that were meant to be temporary—tents, igloos, qarmaqs (sod homes)—moving across a landscape they knew well to ensure their survival. In Northern Canada, colonization intensified between 1950 and 1960, when the federal government forcibly relocated Inuit into permanent communities where they lived year-round [32,33,34]. The first housing policy in the Canadian Arctic was implemented in 1959, an era that consolidated the shift from nomadic to sedentary lifestyles, and from subsistence to wage economy [35]. This period coincides with the establishment of the residential schooling system, which is widely recognized to be an institutional practice of “cultural oppression and forced assimilation” of Indigenous Peoples in Canada ([36], p. S16). In the North, residential schools took the form of day schools and small hostels, where children were separated from their families to attend ([37], p. 4). The intent of these schools was to indoctrinate Inuit children into the dominant Euro-Christian Canadian culture and prevent the transmission of cultural values and identity from one generation to the next ([38], p. 1).

Despite the evolution of housing policies and housing designs, to this day, Inuit housing needs have never been met [1]. Many challenges regarding sustainable housing delivery across Inuit Nunangat persist, including high living and housing costs, a short construction season, and reliance on funding from federal and provincial governments, which does not keep up with the ever-growing housing needs [39]. The resultant housing shortage leads to hidden homelessness and to household overcrowding, since the population has little option but to crowd into available housing stock [40]. In 2016, census data reported that 56.4% of the Nunavut’s population lived in overcrowding, compared to 8.5% of non-Indigenous Canadians [41]. In addition, more than 70% of the communities in Inuit Nunangat do not currently have a safe shelter for women and children experiencing family violence, and where they do exist, they are over-burdened [42]. In two cross-sectional studies conducted in Inuit regions, overcrowding has been associated with increased stress [43] and poor psychosocial health outcomes [44], particularly among women. Baseline data from a rehousing intervention conducted in Nunavut and Nunavik between 2014 and 2017 showed that household overcrowding was associated with a lower sense of home among Inuit adults [45]. The sense of home scale used in this study echoes the notion of “ontological security from the home” in that it assessed people’s perceptions of their home in relation to constructs such as control, privacy, relationships, identity, and safety [45]. Pre- and post-rehousing data of the same intervention research indicated that the reduction in the number of adults per household, and more specifically a transition to nuclear family arrangements (two adults or less with children), were significant predictors of psychological distress reduction [46]. The increase in the sense of home was also a significant contributor to psychological distress improvements. In contrast, longitudinal analyses from other studies conducted in Inuit regions have shown inconclusive results regarding the association between crowding and mental health outcomes [47,48].

### 2.3. Mental Well-Being from an Inuit Perspective

Until recently, the majority of research on mental health among Inuit has used scales or questionnaires developed for, and adapted from, non-Indigenous populations. Grasping the essence of what it means to “be well” across cultures is a challenging endeavour given that every culture has distinct notions of what it is to be a socially valued and well-functioning human being [49]. The Alianait Inuit-Specific Mental Wellness Task Group defines well-being as “*self-esteem and personal dignity flowing from the presence of a harmonious physical, emotional, mental and spiritual wellness and cultural identity*” [50].

In contrast to the emphasis on individuality in much of Western society, the idea of a healthy person in the traditional Inuit culture gives a central role to connections among individuals and to place, a vision that has been called ‘ecocentric’ (as opposed to egocentric) [49,51]. According to Kirmayer (2009), the identity of an Inuk person develops through constant transactions with his or her environment, which includes other human beings, animals, and the land. It follows that healing processes involve, by definition, practices aiming at (re)connecting the people together and with the land, in a way that is comforting, supportive, and productive [51]. One central theme in the literature exploring resilience and mental health from an Inuit perspective is family: spending time with family, talking, sharing food, going out on the land together [49,51,52]. Another central theme in relation to mental wellness is the practice of traditional activities as a means to reconnect with Inuit culture and identity [36,51,53,54,55,56]. Examples of Inuit traditional practices that are meaningful and conducive to well-being include going out on the land, harvesting country food, camping, or arts and craft occupations such as carving and sewing [52]. It must be emphasized that these practices cannot be separated from the family, since traditional skills are learned and enjoyed with family members. Altogether, these insights highlight the relational and holistic ontology of Inuit health.

A recent review of the literature on Inuit wellness identified that research to date has largely been disease-oriented and often emphasizes stigmatizing comparisons between Indigenous and non-Indigenous people. Thus, research that could lead to ways of understanding mental health promotion from an Inuit perspective is needed [57]. In this respect, interventions with positive impacts on family relationships and cultural identity are viewed as more likely to produce benefits, since they address central features of Inuit well-being [36,57,58]. The ontological security theory, as articulated by Dupuis and Thorns (1998) and Kearns et al. (2000), aligns with the Inuit conceptualization of well-being in that it places social relationships and self-identity as foundational processes through which the experiences of home influence health and well-being. It also considers housing functions that intersect with human rights that are considered essential for a healthy development, including the right to access and sustain a safe and stable home [59].

To our knowledge, no research to date has explored Inuit lived experiences of a rehousing intervention. Such knowledge could contribute to “opening the black box” of housing intervention mechanisms to better understand how rehousing leads to mental health benefits, and what circumstances favor or restrain such benefits [60]. Finally, an Inuit-informed understanding of the links between rehousing and mental well-being could be of relevance to territorial and other Inuit organizations who advocate on behalf of Inuit housing and health rights (Section 4.2).

## 3. Research Objective

In the context of the housing shortage that prevails in most communities across Inuit Nunangat, this paper explores the ways in which Inuit adults perceive the rehousing intervention (described below, Section 4.1) to affect—or to potentially affect—their well-being. More specifically, it focuses on the experiences of “home” that are conducive to mental well-being and happiness. It involved Inuit who had been rehoused to a newly constructed public housing unit 1–3 years prior, and Inuit who were still on the waitlist for public housing. The stark contrast between the two housing situations (waitlist versus rehoused) helped to fully capture the extent of housing changes triggered by the rehousing intervention. In line with the Inuit conceptualization of well-being and fundamental concepts in ontological security theory, the rehousing experiences that influenced family relationship and cultural identity are given a particular attention. Ultimately, this study aims to provide a deeper understanding of the ways in which such an intervention operates to promote mental wellness in Inuit regions.

## 4. Research Methodology

### 4.1. The Intervention Studied

The intervention studied in this article is the rehousing of Inuit adults (and their families) into public housing units that were built in several communities across Nunavut following the delivery of a multi-million federal funding plan for public housing [61]. Construction included a mix of one-, two-, and four-bedroom houses and apartments in multiplexes. The study took place in one community where housing needs were considered critical at the time of data collection in 2018 [10] (and still to this day [62]). Available units were allocated on a point-based system, where points varied according to criteria such as household overcrowding, social problems at home, health problems related to housing conditions, number of dependent children, etc. [63]. Each applicant indicated the number of people with whom he/she wanted to move, and this information was considered in the allocation process according to the number of bedrooms required and availability. The wait time for a public housing unit is approximately three to six years across Nunavut [40]. Once tenants have moved in, public housing rent is assessed based on the total gross income(s) of the one or two primary tenants in the unit, and never exceeds 30% of the tenant(s)’ annual income(s). In 2017–2018, 75% of public housing tenants in Nunavut were charged the minimum rent of CAD 60/month since they earned less than CAD 27,041 annually [10].

### 4.2. Research Approach, Sampling, and Recruitment

The active implication of Inuit organizations in housing research is crucial to ensure that objectives are tailored to Inuit-specific housing needs and to optimize knowledge translation strategies [13]. The project *Housing, health and well-being in Nunavut and Nunavik*, in which this study is nested, was developed in collaboration with Nunavut and Nunavik-based organizations concerned with the provision and management of affordable housing, the deployment of public health and social services, and the promotion of Inuit rights and interests (see Acknowledgements) [45]. Partner organizations contributed to developing the objectives and the design of the study. A previous version of the manuscript was shared with regional partners for their critical review and approval, and to ensure that findings were first made available to Nunavik and Nunavut. Three virtual meetings were organized to present and discuss the main findings with partners and other organizations in Nunavut and Nunavik. Several representatives of these organizations, as well as community members, attended these meetings. All comments and suggestions received were integrated into the present manuscript which was again shared with partner organizations, who approved the final version and its submission to the journal.

Data collection for this project was conducted in one Nunavut community, whose name, geographic location, and demographic characteristics are not reported to ensure participants’ confidentiality. The community was selected based on local support to the project, and on the number of public housing units constructed in recent years, potentially affording a reasonable number of participants who moved into these units to be recruited.

Data collection took place over five weeks, from early September to mid-October 2018, using purposeful and snowball sampling methods. Three research interpreters were hired and provided cross-cultural communication insights and invaluable support to participant recruitment. Based on previous field work and interpreters’ experiences, the following recruitment strategies were put in place: door-to-door, local radio announcements, and contacts through the local housing association. Participants who completed the first arm of the project and consented to be contacted for the qualitative interviews were also approached (*n* = 5). Participants either expressed interest directly (face-to-face) or were invited to contact the interviewer or the interpreters to arrange interviews.

Participants were included in the study if they were Inuit adults who had moved to a newly built or renovated public housing unit one to three years prior, or Inuit adults who were on the waiting list for public housing. For the former participants, a window of 1–3 years after rehousing was shown to be sufficient for adaptation processes to take effect and to elicit health benefits [64], while allowing some recruitment flexibility.

### 4.3. Study Design, Data Collection, and Sample Characteristics

The semi-structured interview guide was developed based on housing-related concepts of ontological security initially described by Dupuis and Thorns (1998), and later refined by Kearns et al. (2000) and Padgett (2007): constancy, security, control, identity, and family life. The first part of the interview focused on participants’ experience of these concepts in relation to their current housing circumstances, or in comparison to prior housing conditions for participants who were rehoused. The second part explored perceived impacts on mental well-being more specifically. Links with mental well-being were examined from an Inuit perspective, based on central features of Inuit well-being and happiness: family relationships and cultural practices [50,51,52]. The qualitative interview guide was critically reviewed by northerners (see Acknowledgments) and was adapted over the course of the data collection process.

On average, interviews took between 45 and 60 min to complete. The first author conducted all interviews, with the support of the interpreters for the first 10 interviews. Three out of these 10 interviews served as pilot interviews and were not included in the analysis. Most interviews were conducted entirely in English, except for three interviews during which the interpreters translated some parts of the conversation. The following 18 interviews were conducted without the interpreter to allow participants to express themselves more freely on sensitive or emotional topics. In our view, the gains in truth and profoundness obtained when participants shared their stories unreservedly were worth the slight loss in fluency due to the interpreter’s absence. This dilemma has been observed in cross-cultural community-based research, especially in close-knit communities [65]. In total, 25 participants took part in semi-structured interviews that were included in the analysis.

Out of the 25 participants, 8 were men and 17 were women. Participants who were rehoused (*n* = 11; 4 men, 7 women) were in their twenties or early thirties, they all had children and were living in nuclear family arrangements (1–2 parents with children). Participants who were on the waiting list for social housing (*n* = 14; 4 men, 10 women) presented a broader age range, from early twenties to late fifties. They all had children and typically were living in multigenerational and overcrowded households, with their parents, siblings, children, and their siblings’ children, or a combination of these. The four participants on the waitlist who were not living in overcrowded dwellings were experiencing “hidden homelessness”, i.e., they did not have a usual home and alternated between living with family, friends, or in a cabin on the land during warmer months. All participants who were single parents (*n* = 4)—rehoused or not—were women.

Member checking took place in March 2019, but unfortunately had to be suspended after the first three days due to blizzard conditions, where residents were instructed to stay indoors except for emergencies. Nine of the 25 participants were met during the first three days, and they validated our interpretation of the findings. Preliminary findings were also presented to the community via a special local radio session conducted in English, simultaneously translated into Inuktitut by one of the interpreters. Study procedures were reviewed and approved by the Nunavut Research Institute, by the Comité d’éthique de la recherche en santé de l’Université de Montréal, and by McGill University Internal Review Board. All participants provided verbal informed consents and received a CAD 25 gift certificate to the local general store.

### 4.4. Data Analysis

All interview recordings were transcribed verbatim and imported into Dedoose (OSX version 9.0.17). Codes were deductively developed based on the concept of ontological security, and inductively based on a primary reading of the transcripts. A qualitative thematic analysis of the data was conducted using the deductive–inductive codes and authors kept memos of their analytical thoughts while they coded. To ensure rigour, once all 25 interviews had been coded, the first 5 interviews were recoded to account for the progressive refinement of the codes. Themes were identified from the coded data and were then compared and contrasted to identify high-level themes. The analysis process, including the codes, data interpretation, and relationships between themes was discussed with and validated between authors, who all agreed upon the final themes.

## 5. Results

### 5.1. Themes

Three themes captured the ways in which subjective housing experiences affected mental well-being for those who had been relocated to a new home: (1) refuge creation; (2) self-determination and increased control; (3) improved family dynamics and identity repair. Implicit to these themes were the experiences of participants who had not yet been relocated, opposite in almost every way. A sequence was noted in the themes’ chronology, as refuge creation, self-determination, and increased control gave rise to improved family relationship and identity repair. Themes were not mutually exclusive but rather overlapping. The section below presents these themes, supported by excerpts from interviews. Of note, some sections of the participants’ excerpts were removed from the Results section in consideration of the length of the article—for example, when the interviewer consoled participants, got tissues, asked whether the participants wished the recording to be stopped, etc. This may give the impression that the interviewer was indifferent to participants’ stories, but this was not the case.

#### 5.1.1. Refuge Creation

Participants explicitly identified that rehousing allowed them to gain their own physical space, which permitted the creation of a refuge, or a “safe space” where they could be themselves. In contrast, participants on the waiting list experienced major barriers to refuge creation. The idea of a refuge was linked to well-being by almost all participants. Refuge creation involved three subthemes: security of tenure, sense of safety and comfort, and security of being oneself. The most frequent images depicted by participants to describe home as a refuge included a “safe, peaceful, comfortable, welcoming place”, as for the following participant, who was rehoused the year prior.

Participant 4. Woman, rehoused.
*I guess it’s a safe place for my family, like when I’m at home, […] it’s peaceful and calm and just where I can do anything. It’s like, I don’t know. I guess in a way it’s a welcoming feeling just to be home.* […] *It’s just being at home makes me happy […] to be home, you’re like, you just breathe out and say ‘I’m home’, I can be myself, I can say whatever I want to people around me…*

Family was an integral part of the process of refuge creation, often described as “a place for our own family”, referring to their immediate family. Offering a warm and comfortable place for their children was frequently raised as one of the major benefits of rehousing, one that was linked to happiness.

Participant 24. Man, rehoused.
-INTERVIEWER: … would you say that you are happier now that you moved to this new place or it’s pretty much the same as before?-*Much happier.*-INTERVIEWER: What would be the number one reason for you being happier?-*Giving my children more space, to stay in a warm place. … In our own space.*

The idea of a refuge for those who were rehoused was closely linked to stability and security of tenure, as opposed to participants on the waitlist, who often lived in multigenerational and overcrowded households and needed to move around from place to place when tensions escalated.

Participant 2. Man, rehoused.
*When I used to live in this small unit* [before rehousing], *sometimes we were told to leave that small unit, and then now that I’m home, no one would tell me to leave or anything, so that’s a big difference for me, so.*

Conversely, experiences of hidden homelessness, housing instability, and evictions (or fear of) were reported by participants who did not have their own place. The following participants’ stories are striking examples of how the housing shortage, coupled to the lack of housing options along the continuum (shelter, healing facilities/transitional housing, adapted housing, etc.), creates severe housing instability, suffering, and is an important barrier to refuge creation.

Participant 17. Woman, waitlist, currently living with her child, who had anger issues and broke down her previous unit.
-*When he* [her son] *ruined it* [her first house], *I had to move to his house. […] And they sent him to [place in the south, for anger issues therapy]. So, I stayed where he was staying at. And I was told that my mother was coming back, so the housing had to move me to a place with a rail. And like I said before, I couldn’t do everything for my mom because there’s no handle or anything. So, I brought my mom back to [another community]. And my son came back two months ago. So, they had to move me to another place* [with him], *but with only one bedroom. … It’s like every year I’ve been moving to another place. And I thought everything would be okay now, but I think he still has anger in there (starts crying) … And it’s scary. Especially being alone. Like I don’t have anybody. Just myself… But another thing, like my mom, I miss my mother. And I know she needs me…*

[pause of the recording]
-INTERVIEWER: Okay, so when you’re not at work, where do you go if you don’t want to go back home? Do you just like walk around the community?-*Wander around, go visit some people. But at night, it’s like where am I going to sleep? Because I am scared to go home. There should be shelter here. […] I’m even thinking like when I get paid, to get a shack or a cabin where I can go to.*-INTERVIEWER: A cabin, you mean out on the land?-*Yeah…if I’m scared to go home, I would go to the cabin or a shack, where I can be alone.*

All participants interviewed who were on the waiting list expressed the hope of being allocated a unit in order to create a safe place for themselves and their family, as illustrated by a woman who shared her concerns regarding her children’s safety.

Participants 27. Woman, waitlist, living in an overcrowded dwelling.
-*It’s… (starts crying) It’s hard to find a place when one of my brothers kicked us out from my parents’ house. […] And it’s hard to find a place.*

[…]
-INTERVIEWER. How do you imagine your life in a new place?-*My own space. My kids. To play around and stuff like that. Just be myself. Having my own place, like with my little family, you know? […] Sometimes my kids get scared when one of my brothers gets mad (starts crying)*

Not all participants were living in overcrowded homes before rehousing, but because of the housing shortage, some had no choice but to live with extended family members who did not wish to live with them. This situation of “forced” cohabitation often gave rise to tensed family dynamics and feelings of not being welcome anywhere. The following participant evoked the tension of living with one of her relatives:

Participant 22. Woman, waitlist.
*She doesn’t say it exactly with her words, but I could see it in her face. Like other times she won’t even talk to me if I ask her a question, or she’ll just ignore me. She’ll just have nothing to do with me. That’s when I know she needs her space. I just go out.*

Some participants reported that they needed time to adapt to “make themselves at home” after rehousing; they felt lonely at first, especially children, who were used to living with their cousins, aunts and uncles, and grandparents. However, when participants were asked if they and their family wished to return to their previous place, they all answered no, and emphasized that they eventually adapted, and frequently visited their family.

Participant 1. Woman, rehoused
-*I can say, “I’m going home now, mom,” because I feel at home.*-INTERVIEWER: Oh, okay. And what is home to you? What makes a place be your home?-*When we first moved in—it was quiet and then me and my son would sleep together, but sometimes I felt like when I first moved in there, that I’m a visitor, but I got comfortable later on.*

Participant 8. Woman, rehoused
*Yeah, she* [her daughter] *loved it at my mom’s place* [before rehousing], *but she never liked to be there when she sees my younger siblings fight, like she’s terrified. So, I guess she feels a lot safer here because she never witnessed any fighting… I think she misses her little cousins… all summer they were out playing with each other, but she seems lonely after time, like when she’s home. We’ve been keeping her home because she’s sick, so she seems lonely.*

#### 5.1.2. Self-Determination and Increased Control

Participants on the waitlist reported feeling that they generally had little control over their lives and were trapped in disempowering situations. Mothers and young women, particularly, shared concerning stories about being denied the opportunities to take on desired social roles and embark in important life stages—such as living with their children and being a mother to them; or forming a couple and living together—because of their housing situation. The following participant reported living apart from her son because she was experiencing hidden homelessness after being evicted from her previous apartment. At the time of the interview, she was exploring all possible options to add points to her public housing application, in the hope of being allocated one of the next available units and being reunited with her son.

Participant 22. Woman, waitlist.
-*Well, he stays with* [extended family] *as I am couch surfing* […] *I was calling around and asking how else would I get points? They told me if I go to the meetings and let them know about my situation, that I would be able to get points. They told me the mental health nurse can also help with a support letter. So, I made an appointment.*-INTERVIEWER: Okay. And the [other] letter of support from social services, did you actually see the letter?-*I helped her write it. She was asking me questions. Like where am I staying, how many people are living there, and what’s my situation, or issues.* [So I said] *that I’ve been a single mother for some time now. And we had our own place, but without notice, that we had to go. And I told them how my son is staying over at my* [extended family]*’s while I am sleeping over elsewhere. So, they put that in as well.*

The following situations exemplify how the complex interplay between housing, social norms, taboos, and limited financial literacy, can precipitate, or perpetuate, difficult housing situations. The high rate of overcrowding in the community also limits the help that participants can receive from their social support network. For one woman, the housing shortage meant the near impossibility to leave an abusive relationship.

Participant 12. Woman, waitlist.
*I try the social services, I try to find a place, and they help me, and it was good but my husband go get me, “You don’t live there.” […] I was tired of hearing him, divorce, suicide, and all that. He always tells me to get rid of my life. […]*
*Then* [my husband] *told me that, “If you believe God, you have to come back home.” I believed God, then I went back home. I really believed Him. He helps me a lot. When I pray, he’s there for me. He heard God helps me a lot, protecting me so good, yeah. (crying)*
*I just cried, I just cried, I just talked to the sky, “God help me if you’re there. You’ve got my parents; I’ve got nothing. You gave me a husband. He got me out of the house, and where can I go?” (crying) Nowhere, I just got my mom’s grave, my parents’ grave. I just cried and cried. “Oh My God, you’re out of my life now, how come there’s no help to me. Nobody helps me. I need a place.”…*
*Yeah, I’m just waiting to get a house. If he says, “Get out,” yeah, I’ve got a house, I can get out. Yeah, I want… get a new place, get my job back, bring my kids to school.*

For young adults, living in overcrowded homes prevented them from developing romantic relationships, such as the following young woman:

Participant 15. Woman, waitlist, lives in an overcrowded household.
*It is kind of hard because I have a boyfriend from [another community], he’s usually here because we’re together, but not now, we’re in the long-distance relationship because of our housing, because we don’t have our own room and privacy.*

Participant 13. Woman, not eligible for social housing because she has rent arrears that ought to be reimbursed first. She has been making monthly payments for more than a decade now, out of her social assistance cheque. She still has years to go before she can clear her debt entirely, and then apply for social housing.
*Back in 20xx, I receive a payment* [a rent arrears statement] *of [more than ten thousand dollars], I don’t know where it comes from. I’m not working, I’m a single mother, I don’t know where it came from, but the housing told me to get out, so I had to get out with my children, go back* [living] *with my parents for a while. […] With three different families there, I had to get out and look for a house. When it’s springtime I had to go to the cabin, live in the cabin for six to eight months, come back here. I’m tired of doing this so I had to look for a partner that who could support me. (Starts crying) […] Yeah, I don’t want this, but I am living with an elderly man and I don’t like … I need a house.*

One common point raised by participants was the strains that overcrowding puts on day-to-day routines. Of note, Nunavut communities do not have piped water and sewer systems given the presence of permafrost. Drinking water is delivered by truck, stored indoor in water tanks, which implies a limited usage given the limited storage capacity. In overcrowded households, regular chores can easily take up all the water within a day and sometimes half a day, hence the need for implementing strict water-saving strategies before the next delivery. This results in avoiding cooking, cleaning, or, in some cases, not being able to bathe babies as often as one would like and having difficulties controlling skin rashes. Understandably, this constraining environment necessitates constant negotiations with family members, or major compromises linked to limited space, which leads to social tensions, aggression, and sometimes physical fights. Parents and children often need to share one mattress, with obvious consequences for sleep quality. Many couples mentioned moving back and forth between their respective parents’ places while waiting for tensions to fade out. A man recounted spending days in one bedroom to prevent his child from being exposed to poor air quality (many adults were smoking in an overcrowded dwelling that was poorly ventilated). The child had respiratory health problems and had already required Medivac twice (urgent medical assistance) to seek hospital care in the South. The following participants’ experiences capture the overall feeling of powerlessness that many experienced, and the detrimental impact on children’s well-being.

Participants 32. Man, waitlist.
-INTERVIEWER: Can you give me some more examples, like on a day-to-day basis, what would it [having your own place] change in your day-to-day routines?-*There’d be a lot more smiles, that’s for sure. I would spend more time with my kids instead of just trying to run away from problems that I don’t know how to fix, so many things, like too much actually*. *… Just trying to be a normal person is a real challenge when you have all these problems in front of you. You can’t really do anything about them, except you get out of there and let it be.*-INTERVIEWER: And when you say, “it’s hard to be a normal person,” what it is for you to be a normal person? *Family of mine and its own space and the way they are right now, everybody is like a boss to everybody else. There are too many rubbing shoulders and stuff like that. […]*-INTERVIEWER: So, that is why you feel your kids would be happier [if you had your own place]?-*Oh, yes. A lot happier, for sure. Yes, they would be.*

Participant 27. Woman, waitlist.
-INTERVIEWER: Do you have enough space to cook for the whole family?-*I try to buy food for my family and try to cook enough for my family and my kids, sometimes it’s kind of hard to feed them. […] Too many people in my parents’ house. […] They mostly ran out of food, drinks, sometimes my kids don’t eat for days.*

One of the most important changes resulting from rehousing was the capacity to self-determine one’s life. In some instances, rehousing relieved participants from struggles linked to overcrowding, providing them with more freedom of choice, and, in some others, allowed them to take on life opportunities and elevate themselves into social roles that are culturally valued. These experiences were an important part of their well-being.

Participant 25. Woman, rehoused.
*We were engaged at that time and I didn’t really want to pick a date when we were engaged until we got our own place. That was one of my… me pushing Housing for us to get a place faster and at the same time wanting to have more kids. That’s what I said a couple of times and then a year and a half later we got our place, apartment.*

Participant 28. Woman, rehoused.
-*When I was still with my parents my brothers, both of them were sharing a room and my parents and my sister were in, are still in one room, and me with my husband we had to share with my brother, taking turns so it got pretty annoying sharing a room, so I moved out* [to her boyfriend’s parents’ place]. *That’s the problem, overcrowding.*-INTERVIEWER: Okay so you must have been pretty happy when you learned that it was your turn?-*Yeah it felt so much lighter in my own, everything, not depending on using someone else’s- …like what we do at our own place. Yeah … a good space for me. Nobody bothers us… I can cook anything, I can do my laundry anytime, I can take a shower, my house can be a mess [laughs].*

Many participants pointed out specifically that having their own place allowed—or would allow them—to raise their children “their own ways”. Some participants said that their children lacked boundaries and were too spoiled when living in large households, as depicted by the following participant:

Participant 8. Woman, rehoused.
*My own space, own privacy and easier for me to raise [child’s name] the way I want to and the way [husband’s name] wants to, so that makes it a lot easier… We discipline [the child] more, like we’re not trying to be as careful and like grandparents being grandparents, they try to limp us from disciplining her and we don’t spoil her as much, it helps a lot because she listens a lot more now.*

For others, it was the opposite: children were too restrained because of the many rules that overcrowding implied, as was the case for one participant:

Participant 29. Man, rehoused.
-[Speaking of living in their new unit] *It’s like we grounded, like no more: can we do this, can we do that? A place to call home. Not trying to be quiet in the mornings. More open, more…yes, freedom. And watch my actual family grow. Like I’m from my family, but in my own home, I can raise my own family.* […]-INTERVIEWER: Why exactly do you say that you feel that your children are maybe happier here?-*I don’t know, just… Uncle can I play with this? Uncle can I play with that?” Drinks or anything. Here they can just go in the fridge or in the cupboards. They know where their snacks are.*

#### 5.1.3. Improved Family Dynamics and Identity Repair

The two themes presented so far gave rise to processes that improved family dynamics, and therefore, brought about a positive sense of self. Rehousing created opportunities for participants to self-determine the direction of their lives regarding important life stages and social roles, such as being a wife/husband to a loved one, or being a parent to a child and teaching him/her the “Inuit ways” (important attitudes, skills, and values). Indeed, healthy family relationships represent an essential part of the Inuit identity and are a source of pride. One of the most frequent responses received to the question: “What makes you happy in life?” was: “When my kids are happy, I’m happy.” This speaks to the centrality of family relationships in the Inuit conception of happiness and well-being. Many participants also reported that the increased control they gained from rehousing gave them the freedom to arrange their home environment in ways that were more culturally appropriate and that made the practice of culturally valued activities possible. For one participant, rehousing allowed him to carve as much as he wanted. Carving also represented a source of revenue to afford to go out on the land, another practice he loved.

Participant 7. Man, rehoused.
*It’s like, I can do anything I want here.* […] *I do carvings. I can just go out and make a carving with the tools, or use the water more. Like not saving the water for somebody next. Stuff like that.* […] *Yeah, hunting is my life. [I go out] Like, almost every weekend. Only if I have some gas, I go out. If I don’t have gas, I have to carve or work for gas, until I go out again.*

Participant 28. Women, rehoused.
-INTERVIEWER. Okay and do you also sew?-*Yeah. I like—my favourite thing to do too.*-INTERVIEWER: Yeah? Is it—so do you have enough space here to sew as much as you want?-*Yeah I … I make too much mess sometimes but [laughs] I’m going to be getting a sewing table soon to have it here* [pointing at the second bedroom] *…We all sleep in one room.*

As depicted by the following participant, getting married and starting a family were life stages of profound significance from which participants derived pride. Family rituals, such as name’s sake (soul name), are powerful cultural practices connecting family members together in meaningful ways and contributing to cultural identity. Many participants were waiting to have a place of their own before making decisions of cultural relevance. In a way, rehousing allowed people to progress through life, and in doing so, to affirm and reaffirm cultural symbols.

Participant 29. Man, rehoused.
-[Speaking of his father, who passed away]. *Yes, a matter fact we just went to his grave yesterday to go tell him that I’m married now. Any big occasion or any big event, I still go talk to my dad. He’s not dead, he’s sleeping.*-INTERVIEWER: Can you feel him still with you?-*Yes, I still dream about him. My youngest is named after him, so that takes most of the weight off, like more into the heart.*-INTERVIEWER: Yes, I guess so and I mean, he must be proud of you as well.-*Very, because anything I’d accomplish, he’d be very proud. He’d been there.*-INTERVIEWER: What do you think he’s the proudest of, of everything that you’ve accomplished in your life?-*Me getting married. Yes. Because the night before, this is the table where we had our last conversation and he told me, me and my older brother and my uncle, we’re always going to help each other. We didn’t know why he said that and then he told me I’ll have my own family, I’ll get married, I’ll be working. “No, I want to live with you forever.” He gave me a big hug.*-INTERVIEWER: And then he passed away?-*The next morning.*

Communication was described by participants as one aspect of family dynamics that improved after rehousing, and not only for participants who moved, but for the entire family. The new home represented a space for the rest of the family to go when tensions were escalating.

Participant 4. Woman, rehoused.
*I guess it improved the way we* [extended family] *interact because if they’re having a bad day or like not all the time we’re going to get along with our siblings or parents, so I guess they just come more often when something is happening at their home, so anyway I just, it brings us close.*

Some participants shared their experiences of particularly difficult events and felt that having their own place did, or would, help them recover from their trauma. Traumatic experiences that were mentioned included the suicide of a close relative, sexual abuse, domestic violence, and being forcibly separated from their children. Rehousing was—or would be—a necessary condition for them to start their healing process and to redefine themselves by nurturing healthy family relationships, being a role model for their loved ones, or reconnecting with the culture. The following participant was dealing with anxiety attacks linked to a traumatic event. When asked about what were the things that helped her the most to overcome the difficult situation, she said it was the moments of real connections she now could have with her family.

Participant 28. Woman, rehoused.
*I am able to talk to my daughter now about what she does, what she did and her problems and teach her about what my mom taught me; yeah just talk to her, only her, yeah without anyone around … and also with my husband, good privacy. […] That’s the big help for having your own place we can talk without being disturbed. We don’t have to be in a room or hide or be out to talk. That helps a lot.*

Participants 22. Woman, waitlist.
-INTERVIEWER: […] do you think that having your kids back with you in your own place, do you think that that would stop you from drinking, or help you a little bit?-*Yes. I think it would help me a lot.*-INTERVIEWER: Yeah? […] why exactly?-*I have to look out for them, and I don’t want to let them see me like that because, you know… Having the kids with me would probably stop me from drinking too much.*

Even if rehousing was generally beneficial due to the opportunities it provided, the construction of a cultural identity remained limited in some ways by the cultural inadequacy of the new public housing units. Some mentioned, for example, that the units were not designed to accommodate Inuit families, which tend to be larger than southern families. Many participants who were rehoused were already in a bedroom deficit immediately after moving and mentioned missing space for cultural practices or to simply accomplish daily chores without hassles. Being in a bedroom deficit would not be so much of a problem if there were other easily accessible housing options in the community. Many expressed concerns regarding their children’s future; they were worried about where they would live when they became young adults and wanted to start their lives. The participant cited below expressed mixed feelings; on one hand, she felt relieved that she was rehoused, but at the same time, she anticipated problems arising in relation to a bedroom deficit. She hoped to move to a three-bedroom home soon, but she knew how improbable it was.

Participant 25. Woman, rehoused.
*No, and then there are some people like me waiting to go into a bigger house. There’s a certain age for our kids to not share a room, like when they become, mostly when they’re a girl and a boy. Because I… I want a three bedroom before she is a teenager, my daughter. My oldest will be nine soon. She* [pointing at her youngest] *just turned three, so I want to move into a three bedroom soon, hopefully soon in a few years, hopefully…*

## 6. Discussion

Our findings demonstrate that a rehousing intervention led to important changes in the ways that families experienced their home, which then shaped their well-being. Three main themes were identified to describe the ways in which subjective housing experiences improved the mental well-being of Inuit adults (and their families) after rehousing: (1) refuge creation; (2) self-determination and increased control; (3) improved family dynamics and identity repair. The rehousing experiences of participants in this study share multiple similarities with the experiences of socially marginalized groups who transitioned into public or supportive housing in non-Indigenous contexts [24,27,66]. Across these studies, common themes include longing for safety, stability, autonomy; participants generally viewed their new homes as havens and as an opportunity to contemplate a future. These conditions were perceived as importantly contributing to mental well-being.

In accordance with the experiences of participants waiting to be rehoused in our study, other studies [67,68,69] also reported that living in substandard housing conditions can lead to stressful and chronic problems, such as the lack of privacy, social tensions (sometimes escalating to violence), and the feeling of being unwelcome or trapped in disempowering situations. Common ways to navigate these difficult situations include constantly moving from place to place, and/or turning to alcohol, drugs, or religion. For many participants in this study, the severe housing shortage in their community—and, therefore, the near impossibility of a short-term solution to their problems—created a sense of helplessness, insecurity, and unworthiness, leading to mental distress.

### 6.1. Material Housing Conditions and Inuit Well-Being

In Nunavut’s context, we argue that changes in the materiality of housing are necessary for meaningful experiences to emerge and positively influence Inuit well-being after rehousing. This echoes the idea developed by Saunders (1990), Kearns et al. (2000), and many others (e.g., [69]), that the meaningful experience of “feeling at home” is intricately linked to the materiality of housing. However, housing intervention studies show that the influence of material aspects of housing on well-being is not universal and may vary according to household composition, individuals’ expectations, and previous material living conditions [17,60,70]. In line with the demographics of the Nunavut’s population [71], all participants interviewed had children. The desire to provide a refuge for their family was expressed in both physical (more space and warmth) and symbolic (peaceful, calm, comfortable being oneself) terms, depicting the intricate links between both dimensions. Our findings relate to the conclusions of a participatory action research work conducted in Nunavut, in which respondents identified reasons that they needed a different or a better home, including inadequate design, the age of the dwelling, overcrowding, and being “unsafe for children” [9].

In coherence with the high rates of overcrowding and the proportion of dwellings in need of major repairs across Inuit Nunangat [41], improvements in material aspects of housing in this study (e.g., physical space, available water, thermal comfort, salubrious conditions), provided the necessary conditions for participants to offer comfort and refuge to their families, to self-determine their lives, and to increase the control they had over their daily routines. These changes, in turn, facilitated positive family dynamics and identity repair. The two latter processes share ontological similarities with the Inuit conception of mental well-being. In fact, it could be argued that improved family dynamics and identity repair actually *are* mental well-being benefits (and not processes leading to), according to the Inuit relational and transactional conception of the self [49,51]. While the profound experience of “feeling at home” (encompassing the above-mentioned processes) appears to be one of the central mechanisms by which rehousing improved mental well-being, our findings imply that ‘feeling at home’ was itself determined in large part by the betterment of material housing conditions.

### 6.2. Gendered Housing Experiences in Nunavut

For one woman in our study, eviction resulted in her unwillingly losing the care of her child, who was temporarily living with extended family members, a situation that has been reported elsewhere in Nunavut and Nunavik [72,73,74]. The substandard housing conditions have not only been linked to informal within-family adoption, but also to the over-representation of Indigenous children in Canada’s child welfare system [75]. Considering that single parenthood is a highly gendered phenomenon in which women are more likely to be the primary caretaker [76], and that Nunavut’s residents have little control over public housing construction and allocation, such situations generate great distress, particularly for single mothers. Therefore, increased investments in appropriate, safe, and affordable housing across Inuit Nunangat would significantly contribute to women and children’s well-being. The gender-differentiated experiences of the home have been explored in non-Indigenous groups and are explained in most part by the distinctive social roles that men and women assume within families and communities, and by their differentiated levels of implication in domestic life [22,77]. Reports and research conducted among Inuit women support this proposition, since many responsibilities related to family life tend to be assumed by women, including raising children, caregiving for aging parents, providing food, and balancing work and family [78,79]. These roles are sometimes described as fulfilling and gratifying, positively shaping their social identities, but other times experienced as demanding and worrisome, depending on external factors such as housing and financial security, work schedules, and social support [74,79,80].

As exemplified in this study, domestic violence remains a pervasive social reality that actively shapes the lives and well-being of women and children across Inuit Nunangat [42]. Our findings, along with the recommendations formulated by the Pauktuutit Inuit Women of Canada, highlight the urgent need for emergency shelters serving women and children who are evicted from their home, or victims of domestic violence [81]. This intersects with calls for justice from the Final Report of the National Inquiry into Missing and Murdered Indigenous Women and Girls [82]. Healing facilities have also been identified as critical to preventing future violence and to address unresolved and intergenerational trauma stemming from the experience of colonization [42].

### 6.3. Ontological Security from the Home in the Nunavut’s Context

Colonialism has caused a significant loss in “the confidence that Inuit had in the continuity of their self-identity and in the constancy of their social and material environments” [83], which corresponds to Dupuis and Thorn’s (1998) definition of ontological security. In other words, colonialism has caused profound ontological insecurity among Inuit with the consequences observed today on mental health iniquities between Indigenous and non-Indigenous Canadians [84]. Paradoxically, now that the Inuit have adapted to the Western social and material environments imposed on them (including permanent housing), a serious reform of the very policies that has caused such harm could now be a source of ontological security.

Prior to European contact, Inuit did experience hardship, but in a very different form than experienced today. Colonization and the drastic economic transition that ensued have resulted in a significant decline in self-reliance [79]. Poverty in modern times, from an Inuit perspective, is a situation that exists when working-age adults are not able to access wage-based labor and are also unable to maintain land subsistence activities, which represent an essential component of Inuit livelihoods [85]. The economy of Inuit Nunangat is characterized by an uneven distribution of economic opportunities that benefit communities with extraction industry presence, high costs of goods and services, and barriers to traditional livelihoods despite the profound connection that the Inuit have with the land [13]. According to the Pauktuutit Inuit Women of Canada, sharing and maintaining strong family ties are central values of the Inuit culture [86]. This translates into frequent visits from family members and the presumption that food will be shared with people in need. In this study, participants who had children to feed had to navigate the moral dilemma of sharing limited food with everybody and leaving their children starving, or finding ways to eat away from everybody else. It was evident that, despite the many mental well-being benefits that emerged from the intervention, rehousing did not resolve all the socioeconomic challenges that the Inuit are facing, and these challenges have co-occurring negative influences on mental well-being.

Inuit regions are in a situation where the housing authorities’ ability to generate revenues from housing rent is severely limited by the high poverty rates, and where developers who might be interested in investing in private housing perceive the investment as too risky [87]. In turn, the lack of a housing market perpetuates reliance on subsidized housing and on federal funding, leaving the population with increasing housing deficits. The 2016 census data indicated that more than 80% of the population in Nunavut lived in non-market, subsidized housing, compared to 13% of all Canadians [88]. These economic pressures translate into the erosion of social support networks due to tensions that arise from the housing shortage and pervasive household overcrowding. Many participants in the present study reported that they could not ask for help from their family or find refuge in their place for a long period of time because they were already overcrowded and dealing with their share of problems. This is particularly problematic considering that strong social support has been identified as a “buffer” to the psychosocial health problems linked to overcrowding [44].

### 6.4. Contributions to the Ontological Security Theory

Recognizing the direct links between these historical, social, and economic housing determinants and the way in which people experience their homes brings new insights to the ontological security theory that are Inuit-specific. These elements may apply to other Indigenous groups internationally, who share the experience of cultural identity oppression, including racism, the loss of language, social disruptions, environmental dispossession, and material deprivation, which often lead to spiritual, emotional, and mental disconnectedness [89]. Providing secure and stable homes in Inuit Nunangat will require profound economic transformations and/or a significant shift in political power dynamics. This implies that the constancy (or instability) of social and material environments—an important component of the ontological security from the home—is largely determined by structural factors. What characterizes the housing history in Inuit homeland is that these structural determinants are not, to this day, entirely *self*-determined [13]. At the individual level, rehousing provided participants with the opportunity to self-determine some foundational aspects of their lives. At the collective level, self-determination in the form of economic and housing self-reliance could redefine housing opportunities and transform the lives of a significant proportion of the population. Finally, our findings also highlight that rehousing gave rise to identity repair, which aligns with the last condition identified by Dupuis and Thorns (1998) for home to be a source of ontological security: “home is a secure base for identity construction”. The word “repair” implicitly evokes healing, which was achieved through routinized gestures conducive to social harmony and cultural identity, an aspect that is particularly relevant in the context of cultural oppression.

The housing experiences recounted in this article also substantiate rights-based perspectives on housing, especially in the context of the recently passed *United Nations Declaration on the Rights of Indigenous Peoples Act* (UNDRIP Act) in Canada [90]. Ontological security from the home makes the connection evident between adequate housing, which is a human right in itself [59,91,92], and other rights recognized by the Canadian Charter of Rights and Freedoms, such as the right to security of the person, personal autonomy, and the protection of cultural identity [93]. The ontological security theory was used in other contexts of human rights violations or environmental crisis to illustrate the consequences of displacement and family fragmentation on psychological distress and developmental problems in children during the Bosnian War [94] and Hurricane Katrina in New Orleans [95]. This echoes a report from the *Commission on the Rights of the Child and Youth Protection* in 2007 in Nunavik (Québec), which revealed the negative impacts of housing gaps and overcrowding on the lives of Inuit children, the care they receive, and the abuse they may suffer [72]. The report also confirms that the housing shortage made it harder to recruit case workers and mental health professionals that could have helped parents and children. These conditions altogether infringed the right of children to receive protection, to preserve their cultural identity, and to develop normally in conditions of freedom and dignity [96,97]. In April 2021, very similar conclusions were reiterated in a special report of the same institution [98].

### 6.5. Limitations

This study focuses on subjective housing experiences and their implications on mental well-being. Research partners involved in the larger project in which this study is nested highlighted the interconnections between mental and physical well-being and the fact that housing conditions are impacting both physical and mental health across Inuit Nunangat. The decision to focus specifically on mental health in this paper was guided by knowledge gaps in the housing literature, especially as it concerns housing experiences among Inuit populations. Discussion will continue with research partners to explore ways of reporting on other housing–health ramifications shared by participants. For example, one point that was raised during interviews was that rehousing allowed participants the possibility to isolate sick children and permitted a better control of infectious diseases. The connection between overcrowding and the spreading of infectious diseases in the North has been recognized for a long time, particularly in relation to tuberculosis [99,100], but the COVID-19 pandemic has shed new light on housing issues. The challenges observed in the South regarding physical distancing, isolation, food insecurity, domestic violence, homelessness and the vulnerability of older adults are all exacerbated in the North by the housing shortage, household overcrowding, infrastructure gaps, and the many socioeconomic issues already mentioned [101]. The increased risks of COVID-19 cases in Inuit communities are further compounded by the limited access to clean water, the lack of health professionals/services, and a high prevalence of chronic diseases [84]. This is an evocative example of the domino effect of the substandard housing conditions, and also of the intricate links between physical health (infectious diseases) and concerns that have direct impacts on mental health (food security, violence, etc.).

## 7. Conclusions

To our knowledge, this study is the first to explore how a rehousing intervention shaped the well-being of Inuit, through their experiences and from their perspective. Another innovative aspect of the present article is the use of the notion of ontological security from the home, which helped demonstrate that housing interventions could also bring about far-reaching benefits, touching on issues that are particularly important to Inuit, such as self-determination, family ties, childcare, and cultural revitalization [13,102]. Altogether, the benefits of rehousing could unburden, at least partly, the already scarce mental health services that are much needed in Inuit communities.

Acknowledging the complexity of housing policies and programs is more likely to support innovation in practice because it allows stakeholders to situate their actions within a complex landscape of social and political interests, and to be reflexive about the many ramifications of their work at the individual and community levels ([103], pp. 40-41). This article reflects the many interactions between the rehousing intervention and the colonial and socioeconomic contexts in Nunavut, sometimes interfering with the beneficial effects stemming from the intervention. Nevertheless, over the past several years, there has been substantial work invested in solutions targeted at improving northern housing and socioeconomic outcomes, including new funding initiatives, such as the National Housing Strategy announced in November 2017 [104]. According to Inuit representatives, in order to build sustainable housing in the North, the focus must shift towards recognizing Inuit self-determination and providing housing options that are culturally appropriate, among other things [13,104]. On the latter point, Inuit participants in this study reported the need for safe shelters and transitional housing to protect and support the most vulnerable members of their communities. They also expressed the need for an increased number of affordable housing units that can accommodate various household sizes, including large families, so that more individuals, couples, and families can enjoy more positive interactions between household members, along with pursuing their life goals and cultural practices. Because these needs intersect with recognized fundamental human rights, we argue that construction initiatives increasing public housing stock and addressing gaps in the housing continuum are urgently needed. Such investments would bring Canada closer to meeting its commitment regarding Indigenous rights to adequate housing, enshrined in articles 21 and 23 of the UNDRIP Act [90]. Despite the enormous challenges that lie ahead, the Inuit-informed evidence presented in this article supports the health-promoting potential of sustainable housing delivery across Inuit Nunangat.

## Figures and Tables

**Figure 1 ijerph-19-06432-f001:**
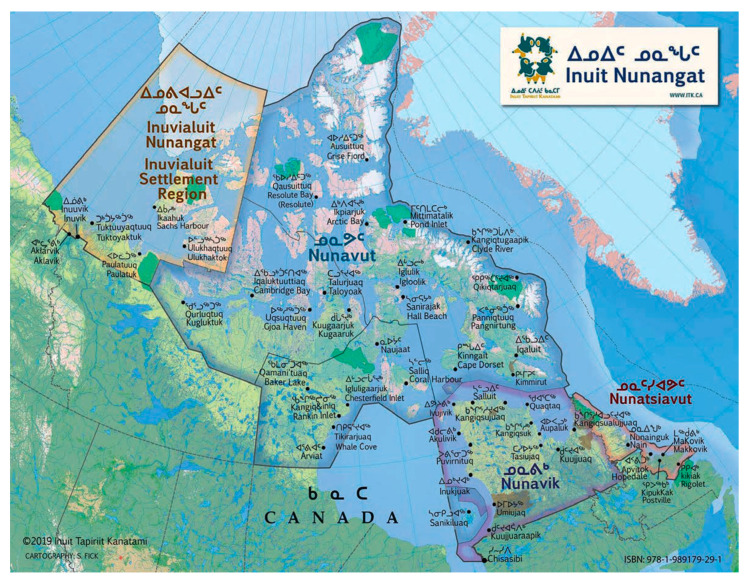
Map of Inuit Nunangat.

## Data Availability

The data presented in this study are available on request from the corresponding author. The data are not publicly available to ensure the confidentiality of participants. Due to the small size of the community where the project was conducted, even the anonymized original data could allow the identification of participants by associations and deductions.

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
