# Peer review of "We’re Home Now”: How a Rehousing Intervention Shapes the Mental Well-Being of Inuit Adults in Nunavut, Canada"

_ijerph, 2022, doi:10.3390/ijerph19116432_

Round 1

Reviewer 1 Report

Interesting topic and well-structured paper

Abstract – words that divide the parts in abstract should be deleted from the abstract (Background, Methods…)

Lines 125-128: there are some value statements – those should be backed by scientific sources or reformulated in more neutral terms

Some referencing is missing in text, this should be corrected

Referencing the authors in text -  such as in Data analysis, “KP developed”… or later in the interviews should be changed to “authors” or “researchers” and “interviewers” for the interview part. At the end of the paper there is a section with the contributions where the authors can write who has done what.

The citations from the interviews should be organized in tables – for example Table X. Extract from the interview… - and then referred to in the text.

Please compare the findings with the results (from existent published research) with effects on other vulnerable groups and single people, or define the gap in it.

Author Response

Dear reviewer,

Thank you for your time and contribution to this paper. Please find my responses to your comments in the document attached.

Best regards,

Reviewer 2 Report

The paper is well written and it was very interesting to read the paper. Please consider the following comments to enhance the quality of the paper.

  1. The authors need to rewrite the abstract in the unstructured format and remove words like Methods, Methods, and Conclusion from the abstract. Please consider looking at a few sample papers previously published in the journal.
  2. The authors need to put more effort into writing the introduction and introducing the problem that is being dealt with in the paper. The authors also need to write more about the research gaps and what way the current paper is attempting to fill in those research gaps.
  3. The author needs to restructure the paper and create a separate section as the Literature Review after the Introduction section. The author should move the sub-sections such as ‘1.1. Housing, mental well-being, and ontological security’ and others to the Literature Review section.
  4. On page 5, it is not clear why the authors have included lines 192 – 202. The findings of the current study should be put in the later sections.
  5. Reconsider moving sub section ‘1.4 The context of the intervention in the present study as the first subsection of the ‘Literature Review. It would give a more logical structure to the Literature Review section.
  6. Reconsider merging the subsection ‘1.6. Research objective’ into the Introduction section.
  7. The authors need to rename section 2. Materials and Methods’ as ‘Research methodology’ section. In management research, it is better to call it that way because no materials are involved in the research as such.
  8. 1 section on sample characteristics on page 7 should be included as a part of the Research methodology section, where the authors could add a sub-section naming ‘Data collection and sample characteristics’.

Good luck!

Author Response

Dear reviewer,

Thank you for your time and contribution to this paper. We carefully considered the comments of the four reviewers; most of them were consistent and were addressed as suggested. However, some of them were conflicting and we tried our best to find the right balance in our responses to fairly address each reviewer’s comments.

Best regards,

Reviewer 3 Report

Thank you for the opportunity to review the article “ ‘We’re Home Now’: How a Rehousing Intervention Shapes the Mental Well-Being of Inuit Adults in Nunavut, Canada”. The article is very interesting and quite engaging, contributing to a new perspective on studying how housing experiences impact the well-being of adults. The authors used a qualitative approach for their study, which is appropriate in such studies when an in-depth study of social phenomena affecting communities is necessary.

Overall, the subject is corrected attributed to the Mental Health section of the International Journal of Environmental Research and Public Health journal “, special issue “Mental Health of Indigenous Peoples” by studying the ways in which a rehousing intervention shapes the mental health of an indigenous community – the Inuits living in Nunavut, Canada.

First, it is important to emphasize that the article is well-structured, the arguments are very clearly presented, and are logically connected with the aim posed at the beginning of the article.

Secondly, the theoretical background and the literature review are appropriate for this subject and for the presented arguments.

I would also like to acknowledge and appreciate the presentation of the context and the state of research in the field, the specification of the limits of the research, and the discussion section.

However, there are some issues that should be addressed in this revision that I consider to be minor.

  • In the Abstract the subtitles should be removed: Methods (line 12), Results (line 16), and Conclusion (line 19).
  • I would recommend that the distribution of the authors' tasks remain specified only as a mention at the end of the article and not within the article. The data analysis procedure (lines 311-321) should be described in a procedural and impersonal way.

Author Response

Dear reviewer,

Thank you for your time and contribution to this paper. Please find my responses to your comments below.

Best regards,

Reviewer 4 Report

The proposed paper is very interesting. 
This study explores the ways in which a rehousing intervention shapes the 
mental well-being of Inuit adults living in Nunavut, Canada.
This article reflects the many interactions between the rehousing intervention and the colonial and socioeconomic context in Nunavut.
But please, complete a few issues:
*Please explain in the paper the extent to which the developed paper can be relevant to the international scientific field.
*In conclusion, please indicate to what extent the proposed paper is innovative and what is its scientific contribution in the field of sustainable housing ? 
* The study lacks an indication of the audience of the data? Who could use the presented study (actors, recipients)? To whom can this knowledge be practical. Maybe it is worth relating it to current events in Ukraine and mass emigration of people in Europe.  
* The study should indicate recommendations in a time horizon. What do the authors think should be the priority in realizing sustainable housing in Nunavut?

Author Response

(The authors gave the same response as above.)
